# An Inflammatory Myopathy in the Dutch Kooiker Dog

**DOI:** 10.3390/ani13091508

**Published:** 2023-04-29

**Authors:** Yvet Opmeer, Guy C. M. Grinwis, G. Diane Shelton, Marco Rosati, Vanessa Alf, Hille Fieten, Peter A. J. Leegwater, Kaspar Matiasek, Paul J. J. Mandigers

**Affiliations:** 1Expertise Centre of Genetics, Department of Clinical Sciences, Faculty of Veterinary Medicine, Utrecht University, 3584 CM Utrecht, The Netherlands; y.opmeer@uu.nl (Y.O.); h.fieten@uu.nl (H.F.); p.a.j.leegwater@uu.nl (P.A.J.L.); 2Department of Biomolecular Health Sciences, Faculty of Veterinary Medicine, Utrecht University, 3584 CL Utrecht, The Netherlands; g.c.m.grinwis@uu.nl; 3Department of Pathology, School of Medicine, University of California, San Diego, CA 94720, USA; gshelton@health.ucsd.edu; 4Section of Clinical and Comparative Neuropathology, Centre for Clinical Veterinary Medicine, Ludwig-Maximilians-Universität Munich, 80539 Munich, Germany; marco.rosati@outlook.de (M.R.); matiasek@patho.vetmed.uni-muenchen.de (K.M.)

**Keywords:** Kooiker dog, myositis, myopathy, dysphagia, dog, autoimmune disease

## Abstract

**Simple Summary:**

We have identified what is most likely a hereditary inflammatory muscle disease within the Dutch dog breed ‘Het Nederlandse Kooikerhondje’. The clinical signs are either locomotory, dysphagia, or a combination of these signs. In most dogs, the muscular enzyme creatine kinase (CK) was elevated. Histopathology revealed an inflammatory myopathy.

**Abstract:**

The Dutch Kooiker dog (het Nederlandse Kooikerhondje) is one of nine Dutch dog breeds. As of 1960, a number of heritable diseases have been noted in this breed. One is an inflammatory myopathy that emerged in 1972, with numbers of affected dogs gradually increasing during the last few decades. The objective of this paper is to describe clinical signs, laboratory results, electromyography and histopathology of the muscle biopsies of the affected dogs. Method: Both retrospectively as well as prospectively affected Kooiker dogs were identified and categorized using a Tiered level of Confidence. Results: In total, 160 Kooiker dogs—40 Tier I, 33 Tier II and 87 Tier III—were included. Clinical signs were (1) locomotory problems, such as inability to walk long distances, difficulty getting up, stiff gait, walking on eggshells; (2) dysphagia signs such as drooling, difficulty eating and/or drinking; or (3) combinations of locomotory and dysphagia signs. CK activities were elevated in all except for one dog. Histopathology revealed a predominant lymphohistiocytic myositis with a usually low and variable number of eosinophils, neutrophils and plasma cells. It is concluded that, within this breed, a most likely heritable inflammatory myopathy occurs. Further studies are needed to classify this inflammatory myopathy, discuss its treatment, and unravel the genetic cause of this disease to eradicate it from this population.

## 1. Introduction

The Kooiker dog, officially called ‘het Nederlands Kooikerhondje’, and hereafter called Kooiker dog, is a small Spaniel-type breed historically used to catch ducks in a decoy [1]. Although the breed most likely dates back to the 16th century [2], it faced extinction in the 20th century as it had lost its purpose. However, shortly after World War II, the breed was re-established by Baroness van Hardenbroek van Ammerstol by carefully selecting a small number of Kooiker dogs. Although she was able to retrieve 30 Kooiker dogs, only 10 dogs were used to re-establish the breed [1,3,4]. The breed gained popularity, and currently it can be found throughout Europe, the USA and Japan. The total number of Kooiker dogs born between 1942 and 2022 is approximately 35,000.

When using such a small number of dogs to re-establish a breed there is an increased risk of genetic disorders. The Dutch breeders association (Vereniging Het Nederlandse Kooikerhondje—VNHK), aware of these risks, promoted the use of all dogs for breeding to limit inbreeding, and health records are kept in a club register that contains information on all the Kooiker dogs born across the globe [3]. Despite this, the first hereditary disease identified in the breed was hereditary necrotizing myelopathy [1]. Cases were seen as of 1962, and eradication of hereditary necrotizing myelopathy by selective breeding only became possible after the discovery of the causative mutation (Mandigers and Leegwater, unpublished data) in 2015. A second genetic disorder emerged around 1994: von Willebrand disease, which again put a strain on the choice of breeding dogs and the genetic diversity of the breed [5,6]. However, because of the implementation of a DNA test in 2000, affected dogs are no longer seen [5,6].

In 1970, the club register made note of a few Kooiker dogs with neuromuscular problems [3]. Although the number of affected dogs remained small during the period of 1970–1990, the number of affected dogs gradually increased hereafter. In 1994, two Kooiker dogs, with clinical signs of exercise intolerance, a stiff gait, cervical ventroflexion, sialorrhea and difficulty eating were referred to the last author (PJJM) and were diagnosed to be suffering from a generalized inflammatory myopathy: (1) symmetric proximal muscle weakness, (2) elevated muscle enzymes, (3) electromyographic readings suggestive for a myopathy, and (4) characteristic muscle biopsy abnormalities.

Inflammatory myopathies are, in veterinary medicine, classified into two categories: idiopathic inflammatory myopathies (IIMs) and secondary inflammatory myopathies associated with other diseases [7,8,9]. Examples of IMs in veterinary medicine include the localized immune-mediated masticatory muscle myositis (MMM) limited to the muscles of mastication [9,10,11], immune-mediated polymyositis (PM) (either focal or multifocal or diffuse) [7,9,12] and canine dermatomyositis (DM), an inflammatory myopathy accompanied by dermatitis and vasculitis. Dermatomyositis has been described specifically in the Collie, Shetland sheepdogs and crossbreds of these breeds [13,14]. In PM, inflammatory cells, including CD8^+^ T cells and macrophages, are concentrated in the endomysium and surround and invade non-necrotic fibers. In DM, the inflammatory cells, mostly B cells and CD4+ T helper cells, are concentrated in the perimysium, where they are either perivascular or scattered diffusely [15].

Inflammatory myopathies are rare diseases in both humans and canines [9]. PM had only been described in a few breeds, such as the Boxer and Newfoundlander, in which sarcolemma-specific autoantibodies have been identified [16,17,18]. More recently, PM has been identified in the Hungarian Vizsla [19,20,21]. Typical clinical signs of dogs affected with PM are exercise intolerance, stiff gait (described as walking on eggshells), muscle atrophy, ventroflexion, dysphagia, sialorrhea and regurgitation. In some breeds, such as the Labrador Retriever, Golden Retriever and Newfoundlander, it appears to affect predominantly appendicular muscles; in the Hungarian Vizsla it predominantly affects masticatory and cricopharyngeal muscles [8,17,20], and in the Boxer it appears to be a combination [17,18,22]. The clinical signs of an inflammatory myopathy are not specific and may be found in other myopathies, such as the various forms of dystrophy. Hence, the actual incidence of myopathies in general or PM specifically may be underestimated in both humans [23] as well as dogs [9].

After the first two cases in 1994, the number of Kooiker dogs with the identified myopathy increased and currently several clinical cases have been identified. The objective of this study is to describe the clinical signs, laboratory results, electromyographic (EMG) findings and histopathology of this inflammatory myopathy in the Dutch Kooiker dog.

## 2. Materials and Methods

The specific aim of this descriptive study was to collect both retrospectively and prospectively Kooiker dogs with clinical signs suggestive of a neuromuscular disorder (myopathy, neuropathy, disorder of neuromuscular transmission).

### 2.1. Study Population

All dogs included in this study were purebred Kooiker dogs. Cases were retrieved (1) retrospectively using the club register of the VHNK [1,3], (2) were made known to us by submission of the medical records, with, if available, blood or pathology samples, or (3) were referred to the last author (PJJM a diplomate of the European College of Veterinary Neurology (ECVN)).

All medical information, either retrieved from the club register or submitted to us personally was obtained with informed consent from the owner. All cases retrieved from the club register that suggested a neuromuscular disorder were evaluated by the last author (PJJM). Cases whose medical records were sent to us could be submitted by the owner, breeder, club, or consulting veterinarian. Of these cases, if available, the pedigree registration number, pedigree name, date of birth (DoB), age of onset (AoO), sex, country of origin, clinical signs, laboratory diagnostics, EMG findings, histopathology and clinical outcome were obtained. If histopathology had not been performed, samples were, if available, sent to the Veterinary Pathology Diagnostic Center of the department of Biomedical Heath Service, faculty of Veterinary Medicine, Utrecht University, the Netherlands (https://www.uu.nl/onderzoek/veterinair-pathologisch-diagnostisch-centrum, or the Neuropathology laboratory, Ludwig-Maximilians-Universität Munich, Germany (http://neuropathologie.de/index.html), for a histopathological examination.

From all cases directly referred, if available, the pedigree registration number, pedigree name, date of birth (DoB), age of onset (AoO), sex, country of origin was obtained after an informed consent. From all Kooiker dogs, a complete history, physical and full neurological examination, including video or MP4 footage, were collected.

### 2.2. Laboratory Measurements

Routine clinical chemistry, haematology and creatine kinase (CK) activity were performed using a freshly obtained serum, EDTA and heparin blood sample. Samples for hematology, clinical chemistry and Toxoplasmosis gondii and Neospora titers were analyzed by the University Veterinary Diagnostic Laboratory (UVDL), University Utrecht, The Netherlands (https://www.uu.nl/uvdl). An additional serum sample was used for the measurement of Myasthenia gravis titers at the Comparative Neuromuscular Laboratory in San Diego, USA (https://vetneuromuscular.ucsd.edu/). As a reference value for CK, 10 to 200 U/L was used.

### 2.3. EMG and Muscle Biopsies

If the clinical signs were suggestive of a neuromuscular disorder as described by Podell (2002) and/or the CK activity was persistently above the reference value of 200 U/L, which suggests a myopathy [24], the dog was anesthetized using generalized anesthesia (Premedication with dexmedetomidine hydrochloride 2 microgram/kg and methadon 0.2 mg/kg intravenously. Induction with propofol 2 mg/kg. After intubation inhalation anesthesia with isoflurane MAC 1.1% and 100% oxygen. After the procedure the dogs were antagonist with atipamezole 20 microgram/kg) and an EMG (Medtronic^®^, Dantec^®^ Keypoint EMG Unit, www.medtronic.com) was performed of the appendicular, axial and masticatory muscles. Two muscle biopsies from the m. triceps brachii (TRI) and m. biceps femoris (BF) of 1 cm by 1 cm by 0.5 cm were obtained and placed in a tube that contained 10% neutral-buffered formalin.

### 2.4. Muscle Histopathology

Muscle samples that were available to us were fixed in 10% neutral-buffered formaldehyde and routinely paraffin-embedded. Sections of 4 µm were stained with hematoxylin and eosin. Distribution and severity of the cellular infiltrations were analyzed by examination in routinely H&E stained sections. For characterization of the inflammatory infiltrate, routine diagnostic immunohistochemistry was performed for ionized calcium binding adapter molecule 1 (IBA1) for macrophages, and cluster of differentiation 3 (CD3) and 20 (CD20) was performed for identification of T and B lymphocytes, respectively. If sample size allowed, the biopsies were sectioned longitudinally and transversely, and were evaluated by a board-certified veterinary pathologist with an expertise in muscle diseases.

The study population consisted of three groups of Kooiker dogs with a possible neuromuscular disease.

Tier I: Kooiker dogs in which the medical record explicitly mentioned suffering from clinical signs suggestive of a myopathy. However, a complete history, clinical data, laboratory examinations, EMG and pathology data are absent, hence these dogs are suspected to suffer from a neuromuscular disease.

Tier II: Kooiker dogs with a history and physical examination suggestive of a neuromuscular disease and a CK activity above the reference value. An EMG may have been performed by the consulting specialist, but a tissue biopsy report is absent.

Tier III: Kooiker dogs in which at least three of the following four criteria published by Bohan and Peter [25] were fulfilled: (1) symmetric proximal muscle weakness, (2) elevated muscle enzymes, (3) spontaneous activity on the EMG and 4) characteristic muscle biopsy abnormalities.

### 2.5. Pedigree Analysis

Pedigree analysis was performed on a subpopulation of Kooiker dogs where common ancestors could be traced back several generations using the club register [3]. A segregation analysis was performed by counts of the patients versus the total count of remaining littermates.

### 2.6. Statistical Analysis

Results were analyzed with IBM SPSS Software, version 28. Descriptive statistics (number of dogs, mean ± standard deviation), chi-squared tests, one-way analysis of variance, and *t*-tests were performed. Results were considered significant when the *p*-value was <0.05.

## 3. Results

### 3.1. Dogs

In total, 183 Kooiker dogs with clinical signs suggestive of (or proven) myopathy were born between 1972 and 2022. Forty dogs were retrieved from the club register or had been made known to us with clinical signs of dysphagia, a possible (tetra)paresis/stiff gait and/or a combination of these clinical signs and reached Tier I level. Another 33 Kooiker dogs were made known to us and reached the Tier II level. Over 87 Kooiker dogs reached the Tier III level of confidence. In 23 Kooiker dogs, the available information was insufficient to classify them into one of the three Tier levels. Hence, the total study population described here consists of 160 Kooiker dogs.

The first 4 cases were born in the period of 1970–1980, the next 3 in the period 1981–1990, 11 in the period 1991–2000, 73 in the period of 2001–2010 and 69 dogs from 2011 until 2022 (Figure 1). Of these 160 dogs, 78 were females, of which 18 were neutered, and 82 males, of which 18 were neutered. In total, 99 dogs were born in the Netherlands, 18 in Germany, 10 in Finland, 6 in Belgium, 5 in Denmark, 6 in Norway and the remaining cases in other European countries, the United States of America and Japan.

### 3.2. Analysis of the Three Levels of Diagnostic Confidence

The clinical signs in affected dogs could be subdivided, as published earlier [8,9,17,22], into three groups: Kooiker dogs suffering from explicitly locomotion problems, dysphagia, or a combination of the two. For 51 dogs, this information was too incomplete to classify them correctly. Classification was therefore performed for 109 dogs. Thirty-nine dogs only had locomotion problems, eight only dysphagia and 62 a combination of both. The age of onset (AoO) was 4 ± 2.3 years (range 6 months to 10 years; 122 dogs) and their age at death (AaD) was 5.5 ± 2.5 (120 dogs). The time between age of onset and age at death (TS) (AaD-AoO) was 1.4 ± 1.7 years (98 dogs). The AoO, AaD, TS and CK activities were, if available, not significantly different for the three Tier groups of confidence (Anova; AoO, *p* = 0.89; AaD, *p* = 0.94; TS, *p* = 0.94; CK, *p* = 0.16) (Table 1).

### 3.3. Clinical Signs

For the analysis of clinical signs, only the dogs with a Tier II and III level were used. The dogs were subdivided into Kooiker dogs suffering from explicitly locomotion problems, dysphagia, or a combination of the two. Thirty-three percent in the Tier II group and 36% in the Tier III group had only locomotion problems (Appendix A). In the Tier II group 19% of the dogs had only dysphagia (Appendix A) and 48% a combination of locomotion and dysphagia versus 3% and 61% in the Tier III group (Appendix A). Dyspnea/panting was seen in up to 36% of all Kooiker dogs (Table 2 and Appendix A). These differences were not statistically significant and therefore all clinical signs, in detail, are reported for the two groups together (Table 2).

Most dogs showed several clinical signs concomitantly. The predominant clinical signs were difficulty walking in 92% of all dogs and a stiff gait in 51% (Appendix A). Up to 40% of the dogs had difficulty drinking and/or eating (Appendix A). Sialorrhea was seen both while eating and drinking in a quarter of all dogs. Nearly half of all dogs developed anorexia. Dyspnea, panting and coughing was present in approximately 20% of all dogs (Table 2).

#### 3.3.1. Laboratory Findings

Routine clinical chemistry and hematology was performed in 110 dogs and did not reveal any significant abnormalities. Serological testing for *Toxoplasma gondii* was performed in 105 dogs, and a minimal rise in IgG (IgG 1:64) for *Toxoplasma gondii* was identified in only two dogs. Serological testing for *Neospora caninum* was performed in 57 dogs and was negative in all cases. In 41 dogs, Myasthenia gravis titers were measured and found to be negative (below the threshold level of the laboratory) in all dogs.

The CK activity was 2056 ± 1601 U/L (107 dogs; range 179 to 8466 U/L) at the time of diagnosis. The CK activity was comparable for the two groups of Tier II and III and was not significantly different (Table 1), nor was there a sex difference for the CK activity (chi-square, *p* = 0.38) or AoO (chi-square, *p* = 0.121), AaD (chi-square, *p* = 0.94) or TS (chi-square, *p* = 0.08). The CK activity was not elevated in one dog (179 U/L), but this dog had received corticosteroids prior to blood collection. However, histopathology of this dog revealed, despite the administered corticosteroids prior to its examination, an inflammatory myopathy.

#### 3.3.2. Additional Diagnostics—Imaging

Thoracic radiographs were evaluated in 27 cases (from Tier II and III). In one dog, a slightly dilated esophagus was noted, and in two dogs a mega-esophagus was identified. Two other dogs showed a broncho–alveolar pattern suggestive of a bronchopneumonia. In all other dogs the radiographic findings were not abnormal.

#### 3.3.3. Additional Diagnostics—EMG’s

EMG examinations were performed on several representative muscles in 90 dogs. Abnormal spontaneous activity was identified in 82 dogs (91%), and electrical silence in 8 dogs (9%). The EMG readings varied from minor abnormalities in 49 dogs, such as a prolonged insertion activity, minimal myotonic discharges and some fibrillation potentials with positive sharp waves, to clear abnormal readings in 33 dogs of myotonic discharges, fibrillation potentials, positive sharp waves, prolonged insertion activity. AoO, TS, AaD, clinical signs or CK activity were not associated with EMG results.

### 3.4. Histopathology

Muscle samples were obtained from 87 dogs, with the histopathology reports from 33 dogs and the actual biopsies submitted from 54 dogs. The triceps brachii and biceps femoris muscles were biopsied the most frequently: 46 samples from the triceps brachii and 38 samples from the biceps femoris. In seven dogs, the masseter muscle was additionally biopsied. In 22 dogs, the exact location of the biopsy was not known. Most cases showed a moderate-to-marked, chronic-active, diffuse, interstitial and myofibre-directed lymphohistiocytic myositis (Table 3 and Figure 2 and Figure 3), typically with a low and variable number of eosinophils, neutrophils and plasma cells (Table 4). In several biopsies, both degeneration (necrosis), regeneration, atrophy and fibrosis were seen. The location of the infiltrate was mainly in the endomysium, or a combination of endomysium and perimysium (Table 3). A varying number of muscle fibers showed hypereosinophilia, loss of cross striations and hyalinization as the hallmarks of myofiber necrosis (Figure 4). Although there was some variation in the extent of inflammation, also between groups of muscle fibers within the same skeletal muscle (Figure 5), it was not correlated with the biopsied muscle, nor with the severity of the clinical signs. The masseter muscle biopsies revealed no clear signs of inflammation. The histiocytic and mixed T- and B-cell lymphocytic character of the infiltrate was confirmed using routine diagnostic immunohistochemistry for IBA1 (histiocytic cells; Figure 6) and CD3 and CD20 (T and B lymphocytes; Figure 7 and Figure 8). 

### 3.5. Treatment/Outcome

All 120 Tier II and III Kooiker dogs were treated with corticosteroids and/or various supplements. The majority of dogs received a dose of 1 mg/kg bodyweight prednisone or prednisolone. Almost all dogs responded well to this treatment at first, but most deteriorated after several months. The treatment of affected Kooiker dogs and the outcome will be described in detail elsewhere.

### 3.6. Pedigree Analysis

Affected Kooiker dogs were derived from 133 litters. In 22 litters more than one dog was affected: there were 18 litters with two affected dogs, 3 litters with three affected dogs and 1 litter with four affected dogs. Of 28 litters, in which 32 affected dogs were born, no information was available on the exact number of littermates, and these were therefore excluded from this analysis. In the remaining 105 litters consisting of a total of 676 dogs, 128 (18.9%) affected Kooiker dogs were born.

## 4. Discussion

This paper describes the occurrence, clinical presentation and histopathological features of an inflammatory myopathy in the Dutch Kooiker dog. In this breed, three clinical presentations are identified, of which locomotion problems with or without dysphagia are the most common. All Kooiker dogs, although seen in several countries, are descendants of 10 founding dogs [3]. Such a small genetic base introduces a great risk for developing genetic disorders [26]. Already, two heritable disorders have been identified in this breed: hereditary necrotizing myelopathy and von Willebrand factor deficiency [1,5]. IM is the third such disorder that may affect this breed [27].

This IM is more common in Kooiker dogs than in most other breeds, and littermates appear to be at increased risk, indicating that it is a heritable disease [27]. As myopathies may be difficult to recognize [8,9], the reported number of 160 is most likely an underrepresentation of the actual number of cases. In this group of Kooiker dogs, some of the dogs showed only minimal clinical signs and were only identified using the measurement of CK and, if elevated, a muscle biopsy. The clinical signs usually start with difficulty in walking accompanied by exercise intolerance. Typical clinical signs include a stiff short-strided gait and/or difficulty getting up. During exercise, affected dogs may show sialorrhea. The measurement of CK activity proved to be very useful in selecting Kooiker dogs with clinical signs suggestive of a myopathy. All Kooiker dogs, even those with only dysphagia as a clinical sign, had elevated CK activities, except for one. Given that CK is not included in most routine clinical chemistry panels, the consulting veterinarian should explicitly request this test. If not carried out, the diagnosis will be missed in a Kooiker dog with the non-specific clinical signs of a neuromuscular disease. As reported earlier, CK activity did not predict the severity of this IM, response to treatment or outcome [28].

An EMG can be a useful tool, although it is not a specific test [29] and it is not widely available in private practice. EMG equipment tends to be expensive and the test is ideally only performed under general anesthesia. Kooiker dogs are reserved towards strangers and may become very defensive when positioned for a blood withdrawal or an awake EMG (personal observations PJJM). For this reason, almost all EMGs were performed under general anesthesia prior to muscle biopsies. As EMG readings can be normal, even in a severely affected patient, histopathology was, in this study, the gold standard.

In people, inflammatory myopathies are a large and heterogeneous group of disorders that have been grouped together collectively. These include polymyositis, adult and juvenile onset dermatomyositis and inclusion body myositis, as well as those of bacterial, parasitic, viral or toxic origin. In addition, there is an IM associated with sarcoidosis and isolated granulomatous myositis. As clinical findings are not specific for polymyositis and can be seen in other generalized myopathies, the clinical findings should not be used to confirm a diagnosis. The diagnosis rests on the histopathology of the muscle biopsy. In 87 Kooiker dogs, we were able to obtain histopathology reports or perform histopathology ourselves. In most Kooiker dogs, differentials such as toxoplasmosis and neospora were excluded. The histopathology is predominantly multifocal and can vary dramatically between muscles. The cellular infiltrate is typically located in the endomysium and/or perimysium with sporadic invasion of non-necrotic fibers. The inflammatory cells are mainly composed of lymphocytes and histiocytes/macrophages. The presence of CD3+ T lymphocytes suggests this may be an immune-mediated PM [8,9,15,17]. However, additional immunohistochemical stainings investigating the exact nature of the infiltrates and possible upregulation of MHC-1 are needed to classify it as such.

No correlation was found between the type of inflammation, the severity of the inflammation and the CK activity or clinical signs. This has been observed earlier in humans with PM [28]. This is similar to other breeds such as the Vizsla, Newfoundlander and Boxer where it is difficult to predict the dog’s prognosis [8,17,20]. As in the earlier study of Tauro et al. [20], a few dogs had a favorable outcome. Clinical signs, CK activity and histopathology were comparable in our Kooiker dogs, which makes it even more difficult to predict the outcome.

Inflammatory myopathies have been described in a few other breeds. In dermatomyositis (ischemic dermatopathy), mainly seen in Shelties and Collies, an interface dermatitis accompanied by a vasculitis is present which is not seen Kooiker dogs [8,13,14]. Polymyositis in the Hungarian Vizsla affects predominantly the muscles of mastication, and only in a limited number are clinical signs of locomotion problems identified [20]. In contrast with the Hungarian Vizsla, where masticatory muscle atrophy is the predominant clinical feature, masticatory muscle atrophy was seen in only 16% of our Kooiker dogs (Table 2). However, histopathology performed on the masseter muscle in seven Kooiker dogs revealed no evidence of inflammation. Despite the fact that almost all Kooiker dogs had locomotion problems, appendicular muscle atrophy was only seen in 7% (Table 2). There is a resemblance with PM seen in other breeds, such as the Labrador retriever and Newfoundlander, but information on PM in these breeds is limited [8,17]. The clinical presentation of these Kooiker dogs is comparable with the Boxer [17,18,22], as well as the type of inflammatory myopathy, but some Boxers were diagnosed at the same time with a neoplasia [17,18], which was not, as far as we know, present in these Kooiker dogs. The Kooiker dog is one of nine Dutch breeds. In 2019, Shelton et al. described an IM in another Dutch breed: the Dutch Shepherd [30]. In the Dutch Shepherd, a mutation in the mitochondrial Aspartate/Glutamate carrier leads to a more oxidizing intramitochondrial environment which results in the IM. The clinical signs of these dogs, all from one family and all of a young age, were generalized muscle weakness and muscle atrophy. The clinical signs are different, as well as the onset; it is not likely that this has the same genetic origin. The Kooiker dog is a spaniel-type dog and as such most likely not closely related to the Dutch Shepherd.

Some points must be raised. First, it is possible that some of the dogs with the confidence activity of Tier I suffer from a different disorder. Inclusion was merely based on the description of the clinical signs. However, these dogs were not used for further analysis and only used for to measure trend in prevalence over time. A second remark is that the medical information of the dogs was sometimes incomplete. Despite the fact that the measurement of CK is easy, it was not always performed. Techniques such as EMG and muscle biopsies may be a financial constraint for an owner, and therefore sometimes lacking. Thirdly, thoracic radiographs had only been made in a few Kooiker dogs. If the clinical signs are indicative for a neuromuscular disease, thoracic radiographs are a worthwhile diagnostic tool in the evaluation of the clinical signs.

The regular occurrence of this IM in siblings indicates that it is an inherited disorder. The accurate identification of affected Kooiker dogs is of utmost importance to enable genetic research. Until a genetic cause has been identified, the situation will remain that a number of Kooiker dogs will be born that will become affected. Future studies are needed to better understand the disease, provide an accurate genetic marker for diagnosis, and improve clinical outcome.

## 5. Conclusions

We have identified an inflammatory myopathy in the Dutch breed ‘Het Nederlandse Kooikerhondje’. The clinical signs are predominantly a locomotion problem with occasional dysphagia. A limited number of Kooiker dogs had clinical signs of dysphagia only. The disease can be diagnosed based on the clinical signs, an elevated CK activity and histologic findings of an IM. Further studies are needed to confirm a diagnosis of immune-mediated PM and to evaluate the possible treatment. As a heritable disease is possible, a molecular genetic study is needed to investigate this disease and eradicate it from this breed.

## Figures and Tables

**Figure 1 animals-13-01508-f001:**
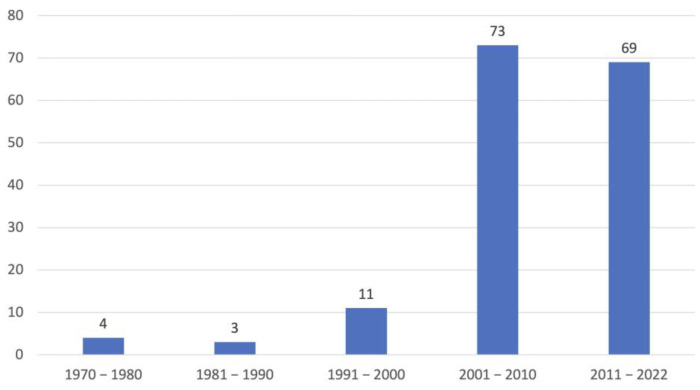
Number of Kooiker dogs identified in the last 5 decades.

**Figure 2 animals-13-01508-f002:**
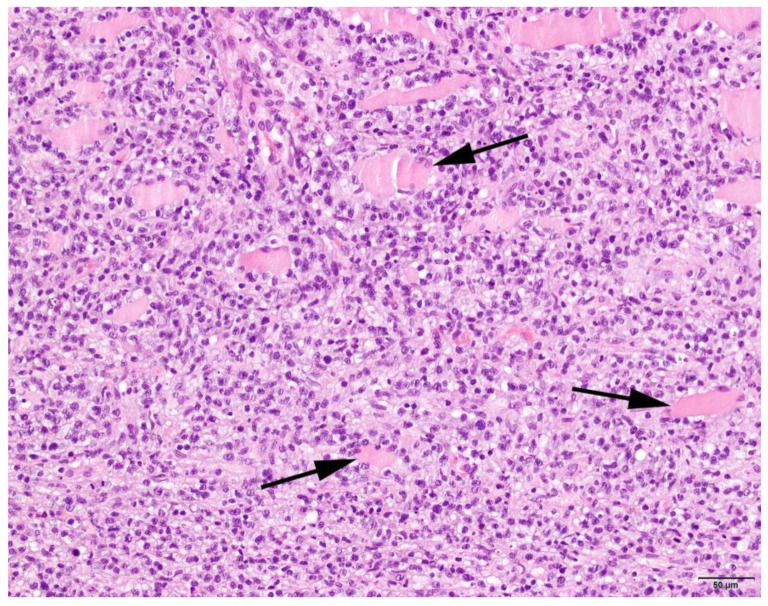
Histology of a skeletal muscle of a Kooiker dog with an extensive mainly lymphohistiocytic myositis with only a limited number of recognizable muscle fibers (arrows). H&E stain, obj. 20×.

**Figure 3 animals-13-01508-f003:**
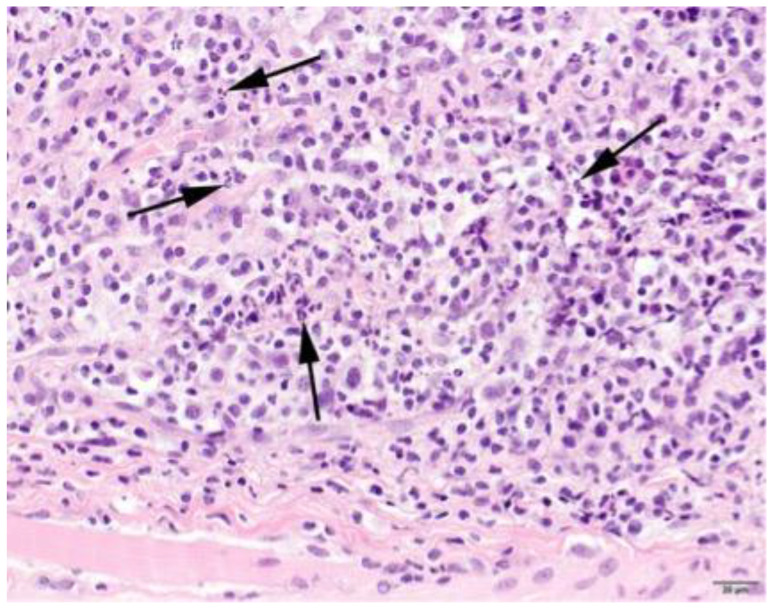
Skeletal muscle of a Kooiker dog with numerous neutrophils (arrows), next to a lymphohistiocytic infiltrate. H&E stain, obj. 40×.

**Figure 4 animals-13-01508-f004:**
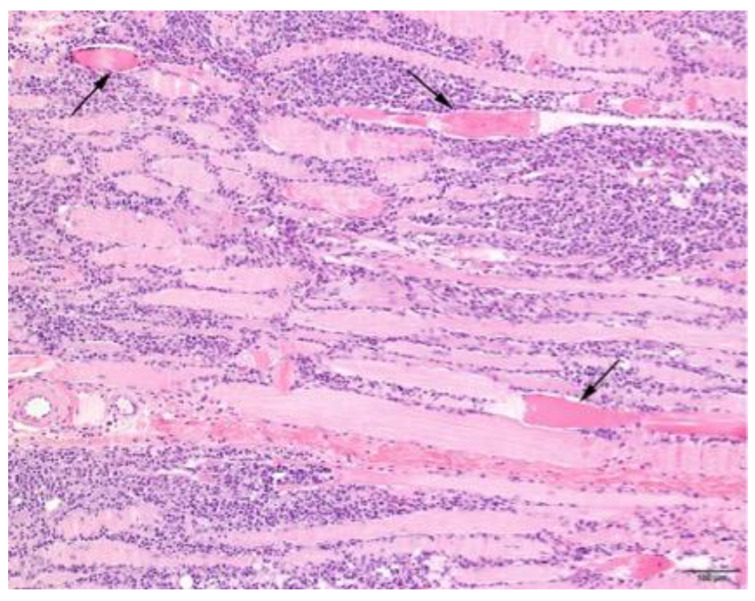
Skeletal muscle of a Kooiker dog with an extensive lymphohistiocytic myositis. Several muscle fibers are hypereosinophilic with loss of cross striation and are sometimes fragmented (arrows). H&E stain, obj. 10×.

**Figure 5 animals-13-01508-f005:**
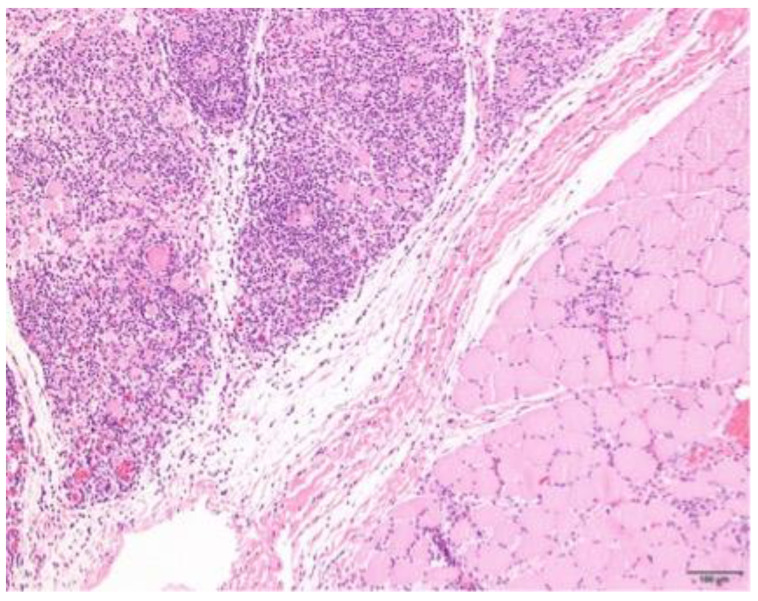
Skeletal muscle of a Kooiker dog with a marked variation in the severity of the myositis (severe upper left side and minimal lower right side) between different muscle fascicles. H&E stain, obj. 10×.

**Figure 6 animals-13-01508-f006:**
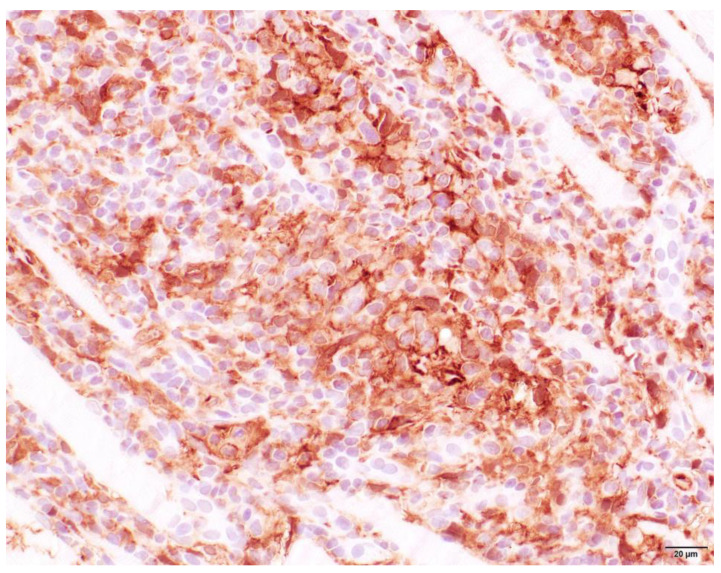
Skeletal muscle of a Kooiker dog with a lymphohistiocytic myositis. Immunohistochemistry reveals numerous mononuclear cells with prominent membranous and mild cytoplasmic immunoreactivity consistent with macrophages (brown cells). IBA1 immunohistochemistry, obj. 20×.

**Figure 7 animals-13-01508-f007:**
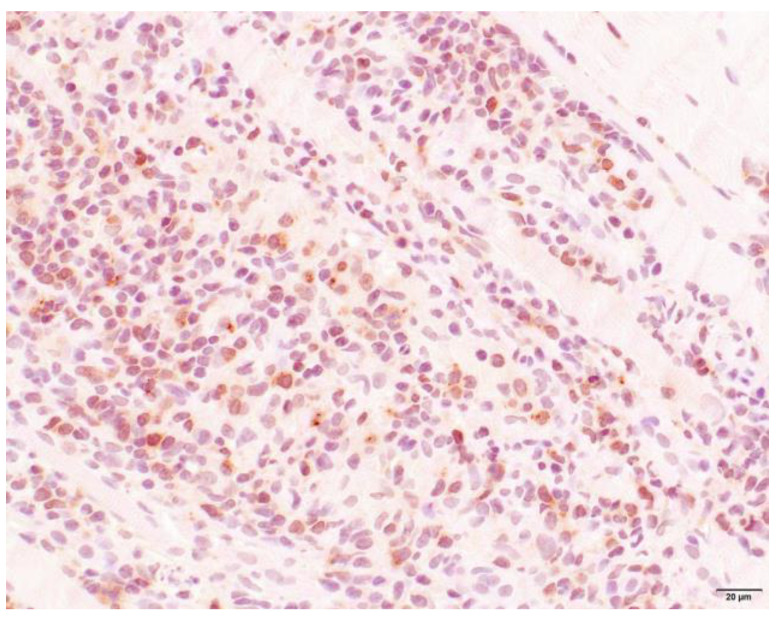
Skeletal muscle of a Kooiker dog with a lymphohistiocytic myositis. Immunohistochemistry reveals numerous mononuclear cells with prominent membranous and mild cytoplasmic immunoreactivity consistent with T-lymphocytes (lightly colored brown cells). CD3 immunohistochemistry, obj. 20×.

**Figure 8 animals-13-01508-f008:**
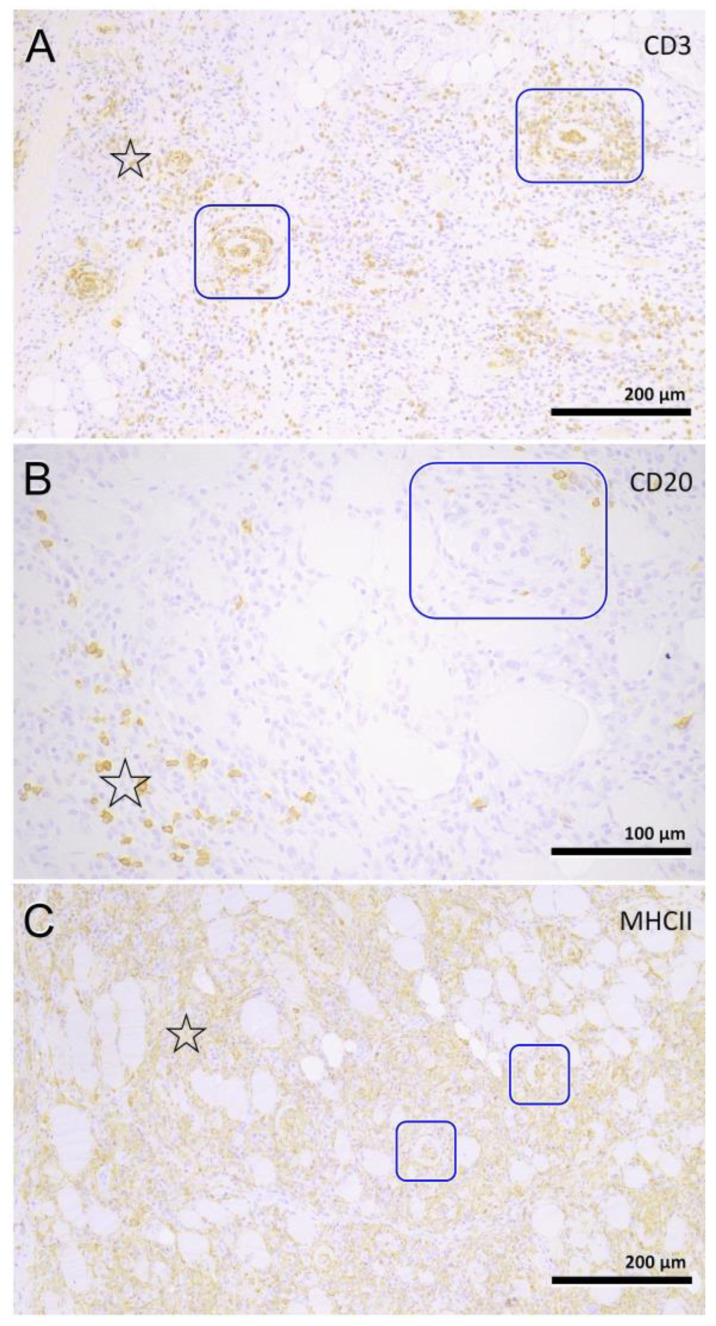
Immune cell phenotyping. Myofiber invasion (frames) is mostly driven by T cells (**A**) and macrophages (**C**), but not by B cells (**B**). All cells are scattered throughout the inflamed interstitial tissue (asterisks). (**A**–**C**): diaminobenzidine, counterstained with hematoxylin-eosin.

**Table 1 animals-13-01508-t001:** For the three diagnostic confidence levels, age of onset, age at death, duration of illness and the CK activity (mean ± SD). There was no significant statistically difference between any of the three groups.

	Tier I		Tier II		Tier III	
		*n*		*n*		*n*
Age of onset (years)	4.1 ± 3.7	10	3.8 ± 1.8	27	4.2 ± 2.2	85
Age at death (years)	5.9 ± 3.1	26	5.5 ± 2.1	24	5.4 ± 2.4	70
Duration of illness (years)	2.5 ± 3.6	8	1.3 ± 1.4	21	1.3 ± 1.4	69
CK activity (U/L)	-		1726 ± 1099	33	2204 ± 1767	87

**Table 2 animals-13-01508-t002:** Clinical signs of all dogs with the level of Tier II and Tier III.

Locomotion		Dysphagia		General	
Difficulty getting up	18%	Difficulty eating	40%	Voice change	12%
Difficulty standing	14%	Unable to swallow food	18%	Dyspnea	22%
Difficulty walking	92%	Gagging while eating/after eating	7%	Panting	36%
Stiff gait	51%	Difficulty drinking	37%	Coughing	17%
Weakness/paresis	18%	Unable to swallow water	17%	Depression	12%
Muscle atrophy limbs	7%	Gagging while/after drinking	4%	Anorexia	49%
Muscle atrophy head	16%	Drooling all day	23%	Fever	5%
Pain while walking	6%	Drooling while walking	8%	Myalgia	14%
		Drooling while eating/drinking	34%		
		Vomiting/regurgitation	6%		

**Table 3 animals-13-01508-t003:** Pattern, expanse, phase and location of the infiltrate in absolute numbers of the 87 muscle biopsies that were available. The information was not complete for all dogs.

Pattern		Expanse		Phase				Location Infiltrate	
Focal	2	Mild	5	Subacute	4	Degeneration	63	Endomysium	27
Multifocal	63	Moderate	13	Chronic	29	Regeneration	38	Endomysium and perimysium	17
		Severe	9			Atrophic	25	Perimysium	5
						Necrosis	31		
						Fibrosis	35		

**Table 4 animals-13-01508-t004:** Type of infiltration and cell type in absolute numbers of the 87 muscle biopsies that were available. + = minimal, ++ = moderate, +++ = severe. The information was not complete for all dogs.

Characterization of Infiltrate and Severity									
Lymphocytes		Neutrophils		Eosinophils		Histiocytes		Plasmacells	
+	18	+	25	+	22	+	16	+	22
++	29	++	10	++	8	++	28	++	14
+++	22	+++	4	+++	6	+++	31	+++	5

## Data Availability

The raw data supporting the conclusions of this article will be made available by the authors, without undue reservation.

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
