# Peer review of "An Inflammatory Myopathy in the Dutch Kooiker Dog"

_animals, 2023, doi:10.3390/ani13091508_

Round 1

Reviewer 1 Report

The paper is interesting. It describes for the first time an inflammatory myopathy that appears to be quite common in a specific Dutch canine breed. A huge amount of clinical data have been collected retrospectively and prospectively. All the sections are well written and sustained with the relevant literature. In particular, histopathologic findings are described in detail. Very useful for the readers are the video showing the main clinical presentations. The interest to the readers would be greater if the paper included the results of genetic investigations and more information regarding response to different treatment strategies.

Minor changes/comments follow:

Line 64: please define the initialism PJJM

Lines 65-68 and lines 162-163: I do not think that it’s correct to directly apply the criteria of Bohan and Peters to our patients. In particular in this disorder considering that some of the dogs had only dysphagia 1) is often non clearly identifiable at neuro examination, maybe easier to establish with EMG. 3) Myopathic changes at EMG is not correct. EMG findings are not specific for a primary muscular disorder. Further,  criteria of Bohan and Peters also included a 5): dermatologic abnormalities.

Line 70: Idiopathic inflammatory myopathies (IIMs)

Line 75: please replace latter with DM or dermatomyositis

Line 163: if you leave this citation please change 1972 with 1975

Line 230: please include values of Toxoplasma serology of these 2 positive cases

Line 233: found to be negative

Line 247: I am concerned about the limited number of dogs that underwent thoracic radiology. Megaesophagus and ab ingestis pneumonia are not uncommon in patients with primary muscular disorders. Considering the prognostic and therapeutic implications of these severe complications, the importance of thoracic radiographs in the work-up of myopathic patients has to be stressed in the discussions.

Lines 255-257: (pseudo)myotonic discharges is an unclear terminology. Please replace, based on the features of the recorded abnormal spontaneous activities, with complex repetitive discharges or myotonic potentials.

Lines 406-407: “The outcome was for most dogs poor but a more extensive study describing  the exact treatment and outcome will be published in another study” please rephrase the sentence.

Author Response

Dear reviewer,

Thank you for your positive comments. We have amended the manuscript based on your suggestions.

The interest to the readers would be greater if the paper included the results of genetic investigations and more information regarding response to different treatment strategies.

A:  Both the treatment as well as our genetic studies would triple the volume of the manuscript. For this reason we have decided to split it into several manuscripts.

Minor changes/comments follow:

Line 64: please define the initialism PJJM

A: done

Lines 65-68 and lines 162-163: I do not think that it’s correct to directly apply the criteria of Bohan and Peters to our patients. In particular in this disorder considering that some of the dogs had only dysphagia 1) is often non clearly identifiable at neuro examination, maybe easier to establish with EMG. 3) Myopathic changes at EMG is not correct. EMG findings are not specific for a primary muscular disorder. Further,  criteria of Bohan and Peters also included a 5): dermatologic abnormalities.

A: You are correct. We have amended the text. The changes are marked yellow.

Line 70: Idiopathic inflammatory myopathies (IIMs)

A: corrected

Line 75: please replace latter with DM or dermatomyositis

A: corrected

Line 163: if you leave this citation please change 1972 with 1975

A: partly deleted, partly corrected.

Line 230: please include values of Toxoplasma serology of these 2 positive cases

A: added

Line 233: found to be negative

A: corrected

Line 247: I am concerned about the limited number of dogs that underwent thoracic radiology. Megaesophagus and ab ingestis pneumonia are not uncommon in patients with primary muscular disorders. Considering the prognostic and therapeutic implications of these severe complications, the importance of thoracic radiographs in the work-up of myopathic patients has to be stressed in the discussions.

A: we fully agree, and this has been added in the 'limitations' part. Our suggestion is to add this to the standard work-up of such a patient.

Lines 255-257: (pseudo)myotonic discharges is an unclear terminology. Please replace, based on the features of the recorded abnormal spontaneous activities, with complex repetitive discharges or myotonic potentials.

A: Not all EMG were made by the same person as several of these dogs were seen by other veterinary neurologists. But you are correct, and the text has been amended.

Lines 406-407: “The outcome was for most dogs poor but a more extensive study describing  the exact treatment and outcome will be published in another study” please rephrase the sentenceOnderkant formulier

A: changed.

Reviewer 2 Report

Comments on the manuscript “An Inflammatory Myopathy in the Dutch Kooikerdog” submitted to the Animals

 General comments

I appreciate the opportunity to review this manuscript.

The study describes the occurrence, clinical presentation, and histopathological features of inflammatory myopathy in the Dutch Kooikerdog. The study was large, including 184 Kooikerdogs dogs. The authors carry out a detailed anamnesis, which aided by clinical and laboratory aspects, conclude that it is a hereditary myositis, whose origin may be in the small number of individuals (n=10) available to recover the almost extinct breed a few decades ago. The authors made differential diagnoses for Toxoplasma gondii and Neospora caninum and did not rule out the possibility of an immune disorder underlying the disease. The manuscript is enriched with videos of the clinical signs of seven dogs. The prognostic of the myositis Kooikerdog is poor; therefore, the authors propose breeding management to extinct that disorder.

The study is an original contribution to clinical approaches and management of Kooikerdog’ myositis. The well-written manuscript provides fundamental knowledge for veterinarians, breeders, owners, and researchers about myositis.

Notwithstanding the well-written manuscript, I make some comments below, which raised questions:

Figure 1: I suggest deleting the title within the figure frame, and inserting in the caption the information that the figure refers to dogs with myositis.

Table 1: Indicate the statistical test used in these analyses. By the way, where are the statistical test results described? It is important for readers to know about these results.

Lines 229-231: The scientific names of the species must be in italics. Please, change "caninium" by caninum

Table 4: The column about lymphocytes is not readable.

Figure 5: Indicate tissue with infiltrate and tissue without infiltrate.

Lines 368 and 393: I assume that there are readers without the excellent ability to interpret histological images. Some readers may learn to interpret histological images with the images. Therefore, I suggest that the images in Figures 6 and 7 indicate the elements described in the captions. 

Author Response

Dear reviewer

Thank you for your review, your comments, and compliments. we have amended the manuscript based on your suggestions.

Figure 1: I suggest deleting the title within the figure frame, and inserting in the caption the information that the figure refers to dogs with myositis.

A: changed

Table 1: Indicate the statistical test used in these analyses. By the way, where are the statistical test results described? It is important for readers to know about these results.

A: the statistical tests are described in M&M. We have added the missing results. Thank you for correcting this.

Lines 229-231: The scientific names of the species must be in italics. Please, change "caninium" by caninum. 

A: Changed

Table 4: The column about lymphocytes is not readable.

A: in our version it was readable. Maybe this has to do with the formatting but I guess it will be corrected by the typesetters in the final version.

Figure 5: Indicate tissue with infiltrate and tissue without infiltrate.

A: corrected

Lines 368 and 393: I assume that there are readers without the excellent ability to interpret histological images. Some readers may learn to interpret histological images with the images. Therefore, I suggest that the images in Figures 6 and 7 indicate the elements described in the captions.

A: it has been added.